# Iron Elution from Iron and Steel Slag Using Bacterial Complex Identified from the Seawater

**DOI:** 10.3390/ma14061477

**Published:** 2021-03-17

**Authors:** Hidenori Tsukidate, Seika Otake, Yugo Kato, Ko Yoshimura, Masafumi Kitatsuji, Etsuro Yoshimura, Michio Suzuki

**Affiliations:** 1Department of Applied Biological Chemistry, Graduate School of Agriculture, The University of Tokyo, 1-1-1 Yayoi, Bunkyo, Tokyo 113-8657, Japan; honhanhenhin345@gmail.com (H.T.); seikaotake123@gmail.com (S.O.); katoh.yugo@gmail.com (Y.K.); yoshimura.sb9.koh@jp.nipponsteel.com (E.Y.); 2Advanced Technology Research Laboratories, Nippon Steel, 20-1 Shintomi, Futtsu, Chiba 293-0011, Japan; etsuroyoshimura@g.ecc.u-tokyo.ac.jp; 3School of Food Industrial Sciences, Miyagi University, 2-2-1 Hatatate, Tauhaku, Sendai, Miyagi 982-0215, Japan; kitatsuj@myu.ac.jp; 4Department of Liberal Arts, The Open University of Japan, 2-11 Wakaba, Mishima-ku, Chiba 261-8586, Japan

**Keywords:** bacterial complex, biofilm, iron and steel slag, iron elution

## Abstract

Iron and steel slag (ISS) is a byproduct of iron refining processes. The lack of iron in seawater can cause barren grounds where algae cannot grow. To improve the barren grounds of the sea, a supply of iron to the seawater is necessary. This study focused on bacteria interacting with ISS and promoting iron elution in seawater. *Sulfitobacter* sp. (TO1A) and *Pseudomonas* sp. (TO1B) were isolated from Tokyo Bay and Sagami Bay. The co-culture of both bacteria promoted more iron elution than individual cultures. After the incubation of both bacteria with ISS, quartz and vaterite appeared on the surface of the ISS. To maintain continuous iron elution from the ISS in the seawater, we also isolated *Pseudoalteromonas* sp. (TO7) that formed a yellow biofilm on the ISS. Iron was eluted by TO1A and TO1B, and biofilm was synthesized by TO7 continuously in the seawater. The present research is expected to contribute to the improvement of ISS usage as a material for the construction of seaweed forests.

## 1. Introduction

Iron and steel slag (ISS) is a byproduct of iron refining processes [1,2,3]. In this research, steel-making slag, which was produced specifically during the steel-making process, was used. The iron content of steel-making slag is higher than that of andesite or cement. Its current usage is limited to cement, pavement, and fertilizer [4]. However, considering the possibility of decreased demand for these materials in the future, new usages are being explored. One of these is the use of its physical and chemical properties as a material for seashore restoration [5]. The previous report showed that the seaweed bed was restored by the treatment with a mixture of ISS and compost. These results suggested that iron elution might be important to promote the growth of seaweed beds.

Iron is an essential trace element for plants, and the lack of iron prevents plants from producing chlorophyll, thereby causing chlorosis [6,7]. In the 1930s, it was found that iron was a limiting factor in biological production in the sea [8]. Martin et al. [9] and others found that the addition of low concentrations of iron to the sea surface resulted in an increase in phytoplankton population [9,10,11]. This result showed that iron limits biological production in the sea.

Recently, barren ground has become a significant problem in Japan [5,12,13]. This phenomenon causes seaweeds to disappear from the seaweed bed, resulting in a decrease in seaweed catch, which consequently negatively impacts the fishing industry. The elevation of sea temperature and grazing by sea urchins are discussed as causes of barren ground; however, the lack of dissolved iron in the seawater might also be a factor [13,14,15]. For example, a sea area with barren ground in Hokkaido, Japan, has an iron concentration of less than 2 nM. Considering that the average iron concentration in the sea is more than 200 nM, 2 nM is extremely low [16]. 

To address this problem, industries related to steel have applied ISS mixed with compost containing humic substances as iron supply materials for the sea. A successful example is that seaweed growth was improved by applying ISS mixed with compost containing humic substances [17]. Although the barren ground was severe before the installation of the mixture of ISS, algae had grown in the area a few months after the mixture was applied [18]. Organic acids eluted from compost containing humic substances in the mixture chelate iron ions and supply to the sea [19]. However, compost containing humic substances should be added periodically for eluting iron chelators.

Hence, another method to efficiently stimulate iron elution from ISS is required. The previous works reported that microbes were used for the bioleaching of ISS indicating that the physiological effect of microbes affect the metal elution of ISS [20]. Some special bacteria also induced the mineralization on the surface of ISS suggesting that bacteria can change the mineral phase of ISS [21]. In this research, microbes that promote iron elution from ISS were examined. Microbes can generate siderophore, organic cherators, and organic acids continuously and grow spontaneously. It is known that there are iron-solubilising bacteria [22] or bacteria inducing iron corrosion [23]. Additionally, microbes that could form biofilm were used to promote better interactions between the microbes and ISS. This is because biofilms interconnect and transiently immobilize biofilm cells, and the biofilm matrix keeps extracellular enzymes close to the cells [24]. The biofilm is a matrix of hydrated extracellular polymeric substances (EPS); the metabolites of microbes [24,25,26], and EPS are mainly polysaccharides, protein, nucleic acids, and lipids [24]. One type of biofilm called the water-channel model has the structure where the biofilm is dotted with microcolonies held in the polysaccharide matrix, and low-density polymers fill the spaces between microcolonies [27]. In the water-channel model, substances are transported through parts with low polymer density. The environment inside the biofilm is different from the outside, and even inside the biofilm the environment differs every µm.

Experiments using ISS as a seaweed-bed-forming material are conducted in sea areas across Japan. However, the effects of the regeneration of seaweed beds vary depending on the area. Effective technology to constantly elute iron from ISS is required. We expect that the results of this research will be useful in projecting ISS as a more valuable material for seaweed bed synthesis through bioengineering using microbes.

## 2. Materials and Methods 

### 2.1. Sample Preparation

Samples of seawater and biofilm on the rocks, concrete, and iron scraps were taken from Tokyo Bay (Figure 1A) and Sagami Bay (Figure 1B), and they were filtrated and spread on Modified Marine Broth 2216. The composition of Modified Marine Broth 2216 is based on that of Marine Broth 2216 which is frequently used to incubate marine microbes. Modified Marine Broth 2216 (Appendix A) was prepared as 1 L solution. The pH was adjusted to 7.6. Samples were incubated at 25 °C for 20 days. 

Carbonated steel-making slag (CSMS) (Figure 1C) was used for the experiments of iron elution. ISS was prepared from converters after optimizing the refining process at Nagoya steel works of Nippon Steel Co., Ltd. (Tokyo, Japan) CSMS is ISS ventilated with carbon dioxide gas, and this was used because its pH was not readily changed even in the water, which offered a suitable environment for microbes. We also used a steel slag hydrated matrix (SSHM) for the screening of bacterial strains synthesizing biofilms. SSHM is made by blast furnace slag, ISS, and water [21]. In this research, CSMS and SSHM were supplied by Nippon Steel Co., Ltd.

### 2.2. Primary Screening to Isolate the Bacterial Strains for Iron Elution by Luminol Reaction

The preculture for each strain was inoculated into Non-Iron Modified Marine Broth 2216 and incubated with 0.5 g CSMS with shaking at a speed of 90 rpm at 25 °C for 3 days. An initial OD_595_ was adjusted to 0.1, and the total amount of medium was 2 mL. As a negative control, 2 mL of sterilized medium was prepared. Thereafter, bacterial cultures were centrifuged with a speed of 5000× *g* for 5 min and supernatants were collected. Non-Iron Modified Marine Broth 2216 (Appendix A) was prepared as a 1 L solution. The pH of was adjusted to 7.6. 

Chemiluminescence intensity in each supernatant was measured by ImageQuant LAS 4010 (GE Healthcare) using luminol reaction. Luminol was oxidized by transition metals including iron and reacted with peroxide iron, which emitted chemiluminescence [22,23,24]. The transition metals concentration is proportional to chemiluminescence intensity. In this research, the iron concentrations in each supernatant were calculated on the basis of linear regression techniques using a series of standard iron solutions (0, 1, 5, 10, 25 ppm).

### 2.3. Identification of Bacterial Strains by PCR and Sequencing

The colony of iron-eluting strain was streaked on a fresh agar plate and incubated at 25 °C for 3 days. Its DNA was extracted using an alkaline heat extraction method. Bacterial 16S rRNA gene primers 10F-5′GTTTGATCCTGGCTCA3′ and 1500R-5′TACCTTGTTACGACTT3′ were used for polymerase chain reaction (PCR) amplification of the 16S rRNA gene. The PCR amplification was performed as follows: each reaction was performed in a final volume of 10 μL containing 0.05 μL TaKaRa Ex Taq, 1 μL 10× Ex Taq Buffer, 0.8 μL dNTP Mixture, 1.0 μL each primer, 1 μL DNA sample, and 5.15 μL deionised water. The reaction mixture was subjected to 35 cycles of amplification as follows: denaturation at 96 °C for 30 s, annealing at 58 °C for 30 s, and extension at 72 °C for 1 min 30 s. The PCR products were purified by ethanol precipitation. The sequence of purified PCR products was analyzed by FASMAC Co., Ltd. (Kanagawa, Japan). The 16S rRNA gene sequences of the top 10 most closely related species were obtained using the Ribosomal Database Project and “Sequence Match” based on type species at NCBI. The genealogical trees showing the relationship between strains acquired by screening and allied species were made by MEGA (Molecular Evolutionary Genetics Analysis). The analysis of genealogical trees was conducted by the neighbor-joining method.

### 2.4. Measurements of Iron Elution Using Inductively Coupled Plasma Atomic Emission Spectroscopy (ICP-AES)

Iron elution by strains TO(Tsukidate and Otake)1A and TO1B were analyzed again using Inductively Coupled Plasma Atomic Emission Spectroscopy (ICP-AES) (SPS3500, HITACHI, Tokyo, Japan). The preculture for 3 conditions: TO1A, TO1B, and TO1, was prepared using Non-Iron Modified Marine Broth 2216. Thereafter each preculture was inoculated into Non-Iron Modified Marine Broth 2216. An initial OD_595_ was adjusted to 0.1, and the total amount of medium was 10 mL. They were incubated with 3.0 g CSMS with shaking at a speed of 90 rpm at 25 °C for 4 days. An axenic condition was prepared as a negative control. Afterwards, bacterial cultures were centrifuged with a speed of 5000× *g* for 5 min and supernatants were collected. Supernatants were filtered using membrane filters of a 0.45 µm pore size, and 0.5 mL of which were added to 0.5 mL of 60% nitric acid (1.38). They were decomposed at 150 °C for 10 h, and the volume of the solution was fixed at 10 mL with 0.1 M nitric acid. Samples were analyzed by ICP-AES under the condition of Fe with a wavelength of 238.277 nm. The iron concentrations were calculated on the basis of linear regression techniques using a series of standard iron solutions.

### 2.5. Scanning Electron Microscopies (SEM) and X-ray Diffraction (XRD) Measurements

To examine the impacts of microbes on ISS, CSMS incubated with TO1A culture, TO1B culture, and TO1 culture were dried and ground to a powder, then investigated by XRD and SEM.

The dried powders were filled up in a nonreflecting silicon sample holder for XRD. XRD patterns were collected using a RINT-Ultima+ diffractometer (RIGAKU, Tokyo, Japan) with CuKα radiation emitted at 40 kV and 30 mA. 

SEM observations were carried out using S-4800 SEM (HITACHI) with a cold field-emission gun at 15 kV. Specimens for SEM were coated with carbon before observations. X-ray microanalysis was performed using energy-dispersive spectroscopy (EDS) with anultra-thin window detector (HORIBA, Kyoto, Japan) equipped to the SEM.

### 2.6. Iron Elution Capacity of Bacteria in the Seawater

A total of 1.0 mL of preculture of TO1 prepared to 0.1 at an initial OD_595_ was inoculated into Non-Iron Modified Marine Broth 2216 with 0.5 g CSMS and incubated for 24 h in 1.5 mL centrifuge tubes. Thereafter, 1 mL of the supernatant was collected, changed to 1 mL of natural seawater, and incubated for 10 days. Since CSMS sank to the bottom of the culture, the supernatant was collected without centrifugation. Seawater samples were taken at 4, 6, 8, and 10-day intervals. Iron concentrations in medium sample and seawater samples were measured and calculated by using luminol reaction. 

### 2.7. Screening of Bacterial Strains for the Synthesis of Biofilm on ISS

To sustain the iron elution of TO1 in the seawater, we expect the bacterial biofilm will cover the surface of ISS and keep the contact of TO1 to ISS. Thereby, bacteria that form a biofilm on the surface of ISS were researched. In this research, SSHM was used as ISS (Appendix A). It suppresses an increase in pH as well as CSMS [25]. As CSMS has a small particle size, it sinks under the water surface despite the property that biofilm is formed near the surface in a short time. We used SSHM that has large particle size for the experiments of biofilm synthesis.

A total of 38 isolates were pre-cultured for 24 h; the preculture was added to Modified Marine Broth 2216, diluted by 0.4 at OD_595_, and cultured with SSHM whose diameter was about 1 cm in 24-well plate at 25 °C for 20 days. During cultivation, the medium in each well was exchanged to 500 μL sterilized medium once a day to prevent excessive pH elevation.

The strain taxonomy was identified by PCR and sequence analysis using the same method that was applied to the identification of strain TO1A and strain TO1B.

### 2.8. Phenol-Sulfuric Acid Method to Quantify the Amount of Biofilm

The amount of biofilm was measured by the phenol sulphuric acid method [26], focusing on the amount of polysaccharide. SSHM was treated by ultrasonic vibration for one minute in small test tubes, then biofilm suspensions were prepared. A total of 0.05 mL 80% (*w/w*) phenol was added, and 0.5 mL concentrated sulphuric acid was added subsequently on the liquid surface. After 10 min, samples were homogenized and put in 25 °C water for 20 min; absorbance at 490 nm was measured.

### 2.9. Iron Elution Capacity of Bacteria with Biofilm in the Seawater

Iron elution capacity of TO1 in the either natural or artificial biofilm was measured by luminol reaction. Natural biofilm was collected from the side wall of the flask in which TO7 was incubated for 5 days, and it was applied on the surface of SSHM. Agarose beads (Appendix A) was used as artificial biofilm and obtained by following these processes. Agarose powder was added to water and boiled, and 100 mL 4% (*w/v*) agarose solution was obtained. A total of 3 mL Tween 80 heated at 47 °C and 50 mL cyclohexane were added to agarose solution, and emulsified for 5 min. Thereafter, 300 mL cyclohexane solution containing 12 mL Span 85 was added there and stirred for 10 min. 

A total of 1.0 mL preculture for TO1 prepared to 0.1 at the initial OD_595_ was inoculated into Non-Iron Modified Marine Broth 2216 and incubated either with 0.5 g CSMS covered with biofilm or with agarose beads for 24 h in 1.5 mL centrifuge tube. Thereafter, 1 mL medium was collected, changed to 1 mL natural seawater (Day 0), and incubated for 10 days. 

Seawater samples were taken at 4, 6, 8, and 10-day intervals. Iron concentrations in the medium sample and seawater samples were measured and calculated by using luminol reaction (Appendix A).

## 3. Results

### 3.1. Primary Screening to Isolate the Bacterial Strains for Iron Elution by Luminol Reaction

Thirty-eight strains were isolated from the seawater and biofilm collected from Tokyo Bay or Sagami Bay. Single colonies on the agar medium were sorted based on appearance, and the collection of bacterial strains was acquired. In this research, CSMS was used as ISS because it inhibits the elevation of pH caused by the treatment that converts calcium hydroxide on the surface to calcium carbonate. One strain that showed the most efficient iron elution was collected from 38 strains using the luminol reaction. In the luminol reaction, luminol was oxidized by transition metals including iron, and the reaction between oxidized luminol and hydrogen peroxide resulted in chemiluminescence (Appendix A). Because chemiluminescence intensity is proportional to iron concentration, the concentration of transition metals can be measured (Appendix A). Using a series of standard iron solutions, it was shown that the chemiluminescence intensity and iron concentration were proportional. The iron concentration was measured based on the intensity. The medium used in these experiments did not include iron ions, unlike the medium used to make the strain library to identify the strain with iron elution capacity in an iron-deficient environment. In addition, the background iron concentration in the medium was the only chemical iron elution which happened equally under any conditions when measuring the iron concentration after the incubation. After three days of incubation for each of the 38 strains in the library and CSMS, the eluted iron concentrations in the medium were measured with luminol reaction based on the chemiluminescence intensity. Strains were named according to the descending order of iron elution from TO1 to TO38 (Figure 2). The relative iron elution by them was also summarized in a table (Appendix A). An axenic condition was prepared as a negative control.

### 3.2. Identification of Bacterial Strains by 16S rRNA Sequencing

Using polymerase chain reaction (PCR) amplification of the 16S rRNA gene, the identification of the strain was attempted; however, the nucleotide sequence of 16S rRNA could not be analyzed because of an overlap of waveform data, indicating the failure to isolate the TO1 colony. Thereafter, the TO1 colony was streaked on a new agar medium, and two colonies with completely different appearances were separated. They were named TO1A and TO1B. PCR amplification of the 16S rRNA gene was performed on these two colonies, and the sequences were analyzed. The 16S rRNA gene sequence analysis showed that strain TO1A and strain TO1B were *Sulfitobacter* sp. and *Pseudomonas* sp., respectively (Figure 3A,B). The closest strain to strain TO1A was *Sulfitobacter pontiacus* (ChLG-10), and the similarity between strain TO1A and *Sulfitobacter pontiacus* [28,29] was 98.9%. In terms of strain TO1B, the closest strain was *Pseudomonas peli* (R-20805)*,* with a similarity of 96.2%.

### 3.3. Chemical Analyses of CSMS after Treatment of TO1

The TO1 (coculture of TO1A and TO1B) culture, TO1A culture, and TO1B culture were incubated with CSMS. The iron concentration in each medium was measured (Figure 4). The control sample did not contain any bacteria. The iron concentration was highest under the incubation with TO1, TO1B, and TO1A in descending order. TO1B contributed more to the iron elution from the CSMS than TO1A. 

Changes in mineral composition after incubation with TO1 culture, TO1A culture, and TO1B culture on CSMS were investigated by X-ray powder diffraction (XRD). In all incubation conditions, the peaks with the strongest intensity of the XRD spectra were attributed to the calcite of calcium carbonate. In addition, peaks attributed to aragonite, magnetite, and hematite were detected. There were no differences in the composition and intensity of peaks between CSMS incubated before and after treatment with sterilized medium, TO1A culture, and TO1B culture, respectively (Appendix A). However, significant differences were found in CSMS incubated with TO1 culture, as compared with the other three conditions (Figure 5A,B). In CSMS treated with TO1, the peak was detected at approximately 26.4°, which is derived from quartz (SiO_2_) {111}. Moreover, by comparing the peaks around 25° to 28° of CSMS incubated with TO1 culture or negative control, CSMS incubated with TO1 culture showed a smaller peak at 27.2° attributed to aragonite {111}, but instead showed a peak at 27.0° attributed to vaterite {112} (Figure 5C). 

Scanning electron microscope analyses of the powdered CSMS analyzed by XRD were conducted (Figure 6). Granular minerals attributed to calcite were found on the surface of CSMS incubated with the sterilized medium as a negative control, while granules with petal-like patterns on the surface were found on CSMS incubated with the TO1 culture.

### 3.4. Iron Elution from CMSM Using TO1 in the Seawater

To confirm the continuous elution of iron from CSMS in the seawater, SSHM was soaked in the TO1 culture medium and then moved to seawater. Iron was eluted until 4 d after moving to the seawater (Figure 7). However, iron elution was almost undetectable after 6 d, and iron elution was not restored after 8 and 10 d. TO1 could not stay on CSMS in the seawater and was spread to the seawater, suggesting the failure of continuous iron elution using the culture medium of TO1.

### 3.5. Screening of Bacterial Strains for the Synthesis of Biofilm on ISS

For continuous iron elution, a biofilm was used to fix TO1 on the surface of the ISS. Biofilm is an organic film made by microbes and sticks to the surface of a solid, which contributes to fixing microbes on ISS. Thirty-eight strains sampled from Tokyo Bay and Sagami Bay were incubated with SSHM for 20 d in 24-well plates. It suppressed the rise in pH, similar to CSMS [30]. After 20 d of incubation, the well plates were observed for the growth of biofilms on SSHMs (Appendix A). Although biofilm cannot be formed well in the other well plates, the wide and broad biofilm was formed in one strain (TO7). The amount of biofilm visually measured are summarized in a table (Appendix A). The picture of well plate revealed that the TO7 strain produced a large amount of yellow biofilm on SSHM (Appendix A). PCR amplification of the 16S rRNA genes and the sequence analysis identified the TO7 strain as *Pseudoalteromonas* sp. (Figure 8). The closest strain to TO7 was *Pseudoalteromonas tunicate* D2, and its similarity to TO7 was 95.8%.

To investigate the amount of biofilm on SSHMs, we developed a phenol–sulfuric acid method to quantify the amounts of saccharide focusing on polysaccharide: one of the main components of the biofilm. In the phenol–sulfuric acid method, the saccharide is measured using the reaction in which the saccharide water solution turns yellow by the addition of phenol and subsequent addition of concentrated sulfuric acid. The maximum absorption wavelength was 490 nm for hexose and 480 nm for pentose and uronic acid. After treatment with the phenol–sulfuric acid method, the quantity of biofilm on the surface of SSHMs incubated with TO7 was measured. Using a series of glucose solutions, it was confirmed that the concentration of glucose and the absorbance of the treated solution at 490 nm had a correlation beforehand. Subsequently, to determine whether biofilm samples correlated with absorbance, the absorbance of dried biofilm was measured at 490 nm (Appendix A). As a result, the quantity of biofilm in the samples and the absorbance were found to be linearly related. 

### 3.6. Iron Elution from CMSM Using TO1 with Biofilms in the Seawater

To fix the cells of TO1 on SSHM, we used both natural biofilm from TO7 and an artificial biofilm made by agarose to compare the effect of biofilms. Artificial biofilms contain air gaps between the networks, and microbes can be alive in the same way as natural biofilms. Either a natural biofilm made by TO7 or the artificial agarose biofilm was mixed with CSMS. Then, TO1 was incubated in each mixture. The iron elution capacity in the seawater was measured for 10 d. The iron concentrations in the supernatants were measured by inductively coupled plasma atomic emission spectroscopy. The supernatants were collected 1 d after the addition of the preculture of TO1, 4, 6, 8, and 10 d after the change of the medium to the seawater. At 1 d after the addition of the preculture of TO1, the iron concentrations were more than 9 ppm in both the natural and artificial biofilms (Figure 9). At 4 d after the medium was changed to natural seawater, there were no significant differences in the concentrations between the two biofilms. After 6 d, iron elution was largely decreased in the condition using the artificial biofilm, while it was decreased by half in the condition using the natural biofilm. After 8 and 10 d, there was no iron elution in the artificial biofilm. On the other hand, iron elution was retained in the natural biofilm.

## 4. Discussion

The technical goal of this research was to identify bacterial strains to promote iron elution from ISS in the sea sustainably. We considered the conditions of the simple culture medium, atmospheric pressure, and temperature between 15 and 25 °C to screen the strains. Although many special bacteria in the extreme environment interact with iron [31,32,33], it would not be good to spread abnormal bacteria that grow in the special medium in terms of ecological preservation. Therefore, Marine Broth 2216 was chosen because its composition was simple, and it was cost effective for screening bacteria cultured on a large scale in the sea. As a result, 38 strains were isolated on Modified Marine Agar 2216. The luminol reaction was applied to measure the amount of eluted iron. The result of Appendix A meant that it was a useful method because the calibration curve with high accuracy was obtained between 1 and 25 ppm. In addition, the luminol reaction does not require any pretreatments, unlike nitric acid decomposition, which makes this screening method rapid and simple.

*Sulfitobacter* sp. (TO1A) and *Pseudomonas* sp. (TO1B) were isolated from the TO1 strain, and TO1B contributed to more iron elution than TO1A. However, iron elution from CSMS was increased when it was incubated with TO1 culture. The reaction between *Pseudomonas* sp. and *Sulfitobacter* sp. is called quorum sensing. For example, the production of the siderophore pyoverdine by *Pseudomonas aeruginosa* is influenced by quorum sensing [34,35]. According to previous research, pyoverdine was produced less by the *P. aeruginosa* strain missing the quorum sensing regulating gene, *lasR*, than wild strains. Conversely, Lewenza et al. [36] revealed that the production of the siderophore, ornibactin, decreased if *Burkholderia cepacia* does not have the *cepR* gene that regulates quorum sensing. Quorum sensing had some impact on iron elution during co-cultivation.

The composition of CSMS incubated with TO1 culture was analyzed using XRD. The unique changes in the spectra were detected only after incubation with TO1 culture, including the appearance of a strong peak at 26.4° that is in accordance with the peak derived from quartz {111}. The original CSMS did not contain quartz, and the major form of silicon was amorphous silicon dioxide. A possible reason for the appearance of quartz after incubation was that silicon dioxide was eluted first and then recrystallized as quartz during the incubation process. The elevation of pH largely contributed to the elution of amorphous silicon dioxide, and the recrystallization of silicon dioxide was caused by the decrease in pH on the surface of the CSMS. The secretion of acidic organic molecules, including iron chelator, led to a decrease in pH, and this phenomenon might happen specifically in the incubation of TO1. The assumption about the formation of quartz was one of the hypotheses from the XRD results. Further analyses should be necessary to reveal the mechanism. The other change was that the peaks of aragonite lowered, and the peaks of vaterite raised due to the incubation of TO1. SEM observations showed petal-like patterns on the surface of CSMS incubated with TO1 culture, which is a unique feature of vaterite. Vaterite crystal tends to make the petal-like patters on the crystal surface because the stability of vaterite crystal is unstable and repeats the precipitation and dissolution at the molecular level. Such unstable phase of vaterite may be a main factor to form the petal-like patterns. It was reported that aragonite was dissolved by the local pH decrease, and vaterite was deposed in the presence of a large quantity of saccharide [37]. The solubility of aragonite or vaterite in the water is different. The change of mineral composition in CSMS indicated that the solubility of mineral components may be changed by the effects of TO1. From this report, it was assumed that the dissolution and recrystallization of CSMS occurred actively during the incubation with TO1 culture.

Since the final goal of this research was to interact ISS and bacteria in the sea to promote iron dissolution, TO1 should grow with sufficient density near the surface of ISS in the sea that does not contain many nutrients. When TO1 culture was incubated with CSMS in seawater, almost all iron elution disappeared after 6 d because cells of TO1 were removed from the surface of CSMS. Then, we added the biofilm to TO1 incubation to CSMS to keep the binding of bacterial cells on the surface of CSMS. 

A number of biofilm products waere observed to search for bacteria producing biofilm on the surface of SSHM. SSHM was used in this experiment because it had a larger particle size and did not increase the pH of the solution much more than CSMS.

Although the crystal violet staining method is generally used to quantify biofilms [38,39,40], it could not be applied to ISS because pigments of crystal violet were absorbed on the surface of ISS, and ISS were dyed with or without biofilm, resulting in the failure of accurate measurements of biofilm. We successfully invented the quantification method for biofilm on the surface of SSHMs by using the phenol–sulfuric acid method.

Iron elution from CSMS was sustained only when it was incubated with a mixture of TO1 and natural biofilm from TO7. EPS composing biofilms seem to contribute to bacterial growth in the natural biofilm [41]. Because it contains various polymeric substances, including saccharides, proteins, and nucleic acids, bacteria can easily absorb necessary nutrients, including carbon, nitrogen, and phosphorus [24,25,26,42]. Besides the nutritional aspect, saccharides in EPS are hydrophilic and charged, and therefore ions can be easily exchanged between the inside and outside of the biofilm. 

Due to these factors, bacteria grew in the natural biofilm, not in the artificial biofilm, and iron elution by bacteria was sustained for a long time by forming biofilms on the surface of CSMS.

## 5. Conclusions

The purpose of this research was to identify the bacteria that interact with ISS and promote iron elution. In 38 strains collected at Tokyo Bay and Sagami Bay, the TO1 strain, which promoted iron elution from ISS was found. TO1 was a mixture of TO1A (*Sulfitobacter* sp.) and TO1B (*Pseudomonas* sp). TO1B primarily contributed to the iron solution. When TO1B was co-cultured with TO1A, iron was eluted more than only TO1B, suggesting that the two bacteria interact with each other to induce iron elution. Quorum sensing may activate the iron chelator biosynthesis pathway, which is not activated by only one strain of bacteria. The culture of TO1 with CSMS changed the composition of the minerals. Quartz appeared, and aragonite became vaterite after the incubation of the TO1 culture with CSMS.

To maintain continuous elution of iron from CSMS, TO1 was cultivated in the biofilm formed by TO7 (*Pseudoalteromonas* sp.). The extracellular polymers in the natural biofilm may be useful for nutrition and stimulate bacterial growth to increase the elution of iron from CSMS. To optimize the combination of bacteria and cultivation conditions with ISS for the materials of seaweed bed synthesis, further research is needed.

## Figures and Tables

**Figure 1 materials-14-01477-f001:**
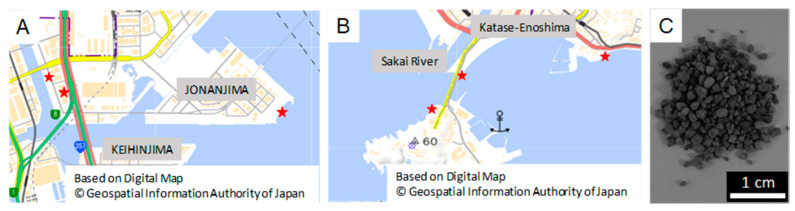
Bacterial sampling and iron and steel slag (ISS). (**A**) Tokyo Bay, where bacteria were collected. (**B**) Sagami Bay, where bacteria were collected. Stars indicated the position of sampling. (**C**) Picture of carbonated steel-making slag (CSMS).

**Figure 2 materials-14-01477-f002:**
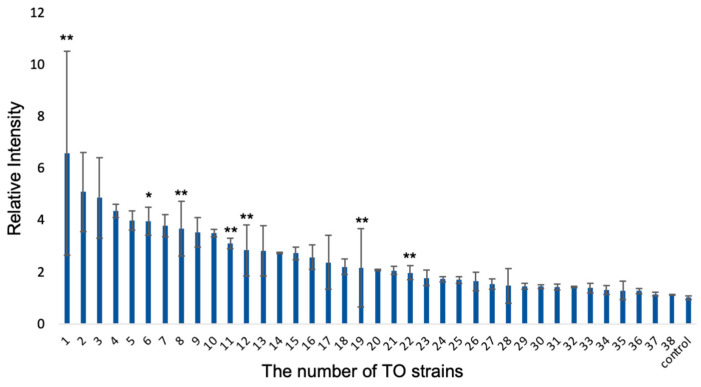
Relative chemiluminescence intensity by luminol reaction for 38 strains in the TO library named in descending order of iron elution from TO1 to TO38. TO1 showed the highest contribution to iron elution. (Mean ± SD, *n* = 3). (Dunnett’s test, * *p* < 0.05, ** *p* < 0.01).

**Figure 3 materials-14-01477-f003:**
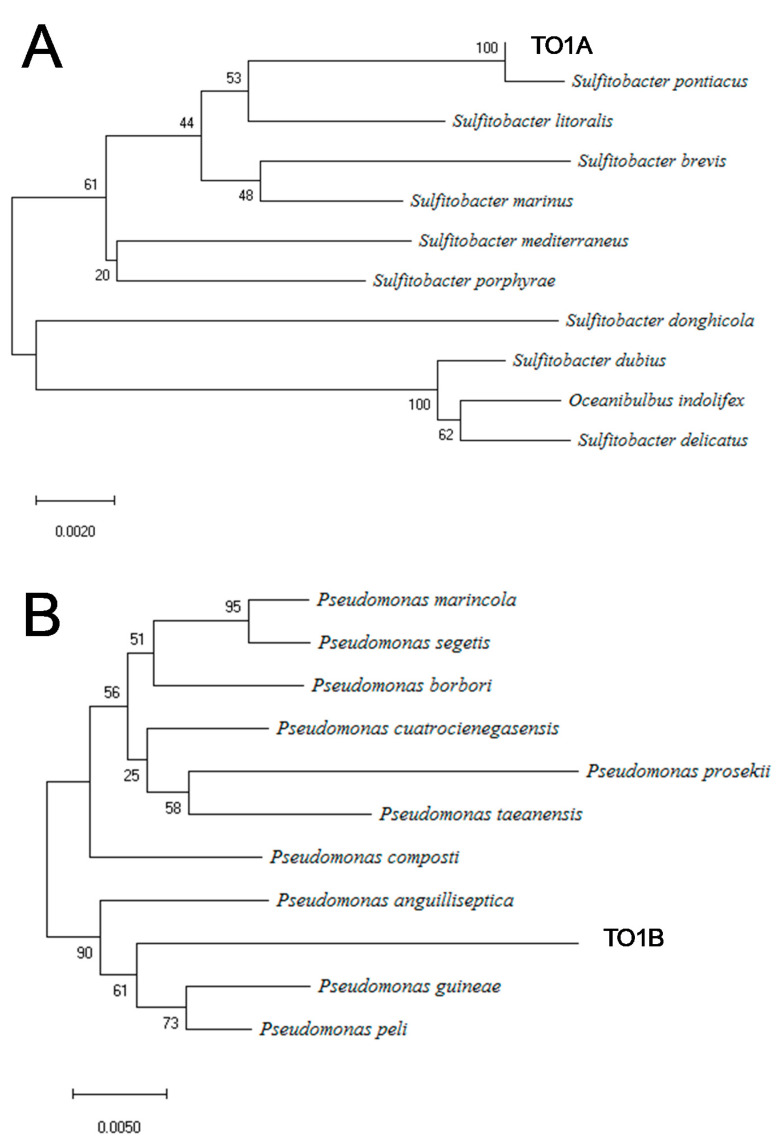
Phylogenetic trees of TO1A and TO1B using 16S rRNA sequences. (**A**) Strain TO1A was identified as *Sulfitobacter* sp. (accession numbers are in Appendix A). (**B**) Strain TO1B was identified as *Pseudomonas* sp. (accession numbers are in Appendix A).

**Figure 4 materials-14-01477-f004:**
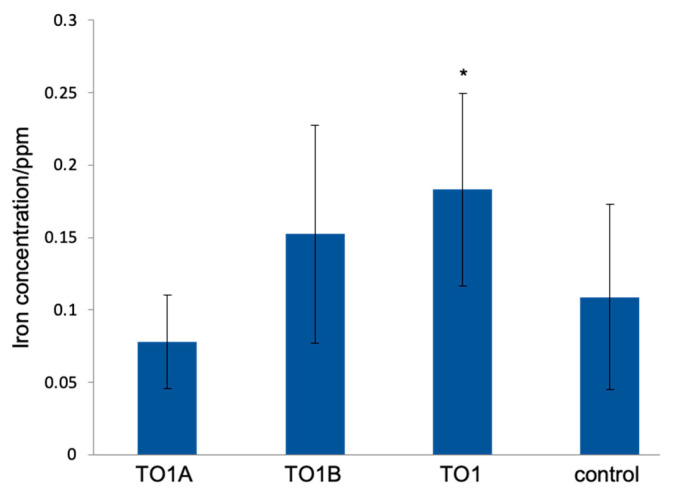
Iron concentrations eluted by the bacteria. CSMS were incubated with TO1 (TO1A and TO1B) culture, only TO1A culture, or only TO1B culture. Iron elution was measured by ICP-AES. (Means ± SD, *n* = 9) (Dunnett’s test, * *p* < 0.05).

**Figure 5 materials-14-01477-f005:**
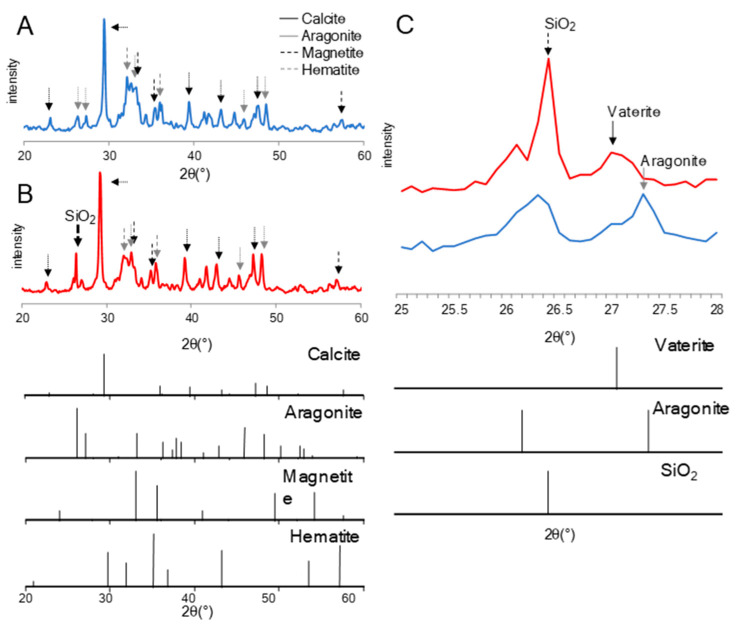
X-ray powder diffraction analysis of CSMS. (**A**) When incubated with sterilized medium, peaks attributed to aragonite, magnetite, and hematite were detected. (**B**) When incubated with TO1 culture, the peak was detected at approximately 26.4° derived from quartz (SiO_2_). (**C**) CSMS incubated with the TO1 culture showed the appearance of a peak at 27.0° attributed to vaterite {112} and showed a decrease in peak intensity at 27.2° attributed to aragonite {111}.

**Figure 6 materials-14-01477-f006:**
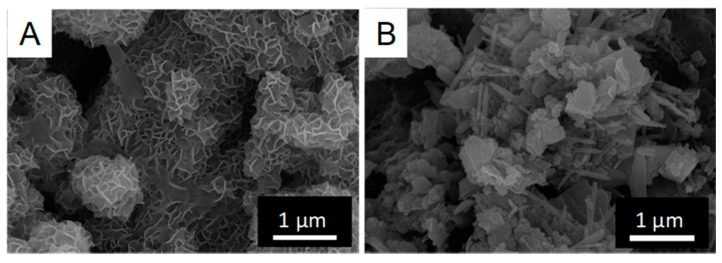
SEM observations of CSMS. (**A**) Granules with petal-like patterns were found on the surface of the CSMS incubated with TO1 culture. (**B**) Granules without petal-like patterns were found on the surface of the CSMS incubated with the sterilized medium.

**Figure 7 materials-14-01477-f007:**
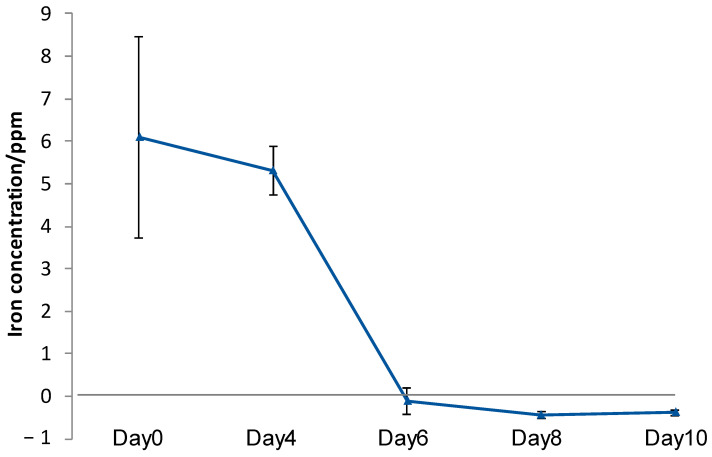
Iron concentrations eluted in medium (day 0) and seawater (day 4, 6, 8, 10). When CSMS was incubated with TO1 culture, iron elution was not detected after 6 d, and it did not restore after 8 and 10 days. (Means ± SD, *n* = 6).

**Figure 8 materials-14-01477-f008:**
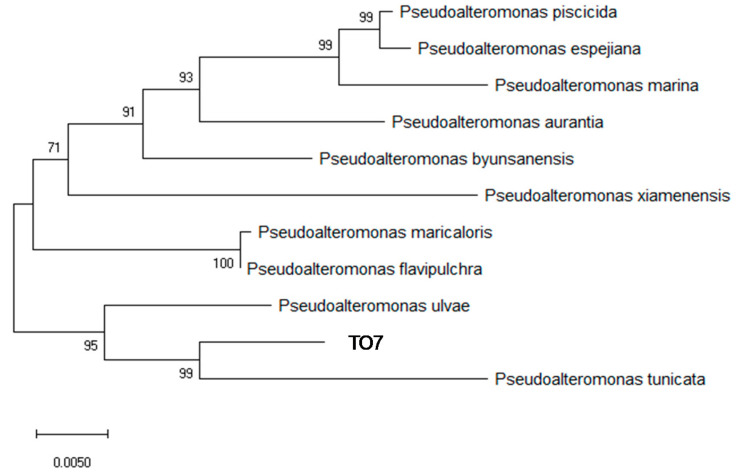
Phylogenetic tree of TO7 using 16S rRNA sequences. TO7 was identified as *Pseudoalteromonas* sp. (Accession numbers are in Appendix A).

**Figure 9 materials-14-01477-f009:**
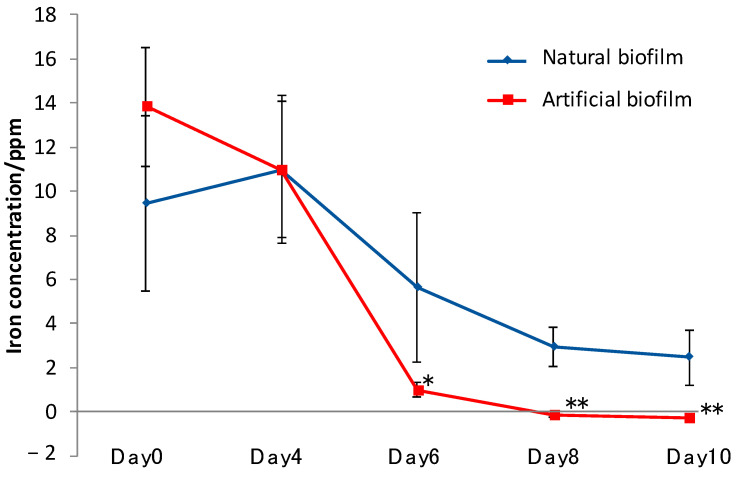
Iron concentrations eluted in medium (day 0) and seawater (day 4, 6, 8, 10) in the presence of biofilm. After day 4, the medium was changed to natural seawater. There were no significant differences in the concentrations between artificial and natural biofilms. After day 6, iron elution was largely decreased in the condition using artificial biofilm as compared to the natural biofilm. (Means ± SD, *n* = 6) (*t*-test, * *p* < 0.05, ** *p* < 0.01).

## Data Availability

The data presented in this study are available on request from the corresponding author. The data related to ISS, CSMS, and SSHM are not publicly available due to industrial cooperation with Nippon Steel Co., Ltd.

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
