# Peer review of "Iron Elution from Iron and Steel Slag Using Bacterial Complex Identified from the Seawater"

_materials, 2021, doi:10.3390/ma14061477_

Round 1
Reviewer 1 Report
This is an interesting study. The author described an approach to improve the steel slag usage as a material for the construction of seaweed forests. There are several concerns need to be addressed before publication.
Major Concerns:
- A proper comparison with related work in the Introduction section is necessary;
- In Figure 2, it was stated that “TO1 showed the highest contribution to iron elution”, but it should be noted that the error bars presented in Figure 2 are significant. Are the presented data statistically different from each other?
- What those marks X, Δ and О stand for in Table S1?
- Proper descriptive phrase for section 3.2 is required
- It seems the iron concentration eluted by TO1A is less than that eluted by the control in Figure 4, why? Similarly, are each group statistically different from each other?
- XRD was utilized to investigate the changes in mineral composition after incubation with TO1 culture, why is this composition change important?
- It is better for the authors to explain the mechanism of petal-like patterns formation in Figure 6;
- Lines 345-346, “At 4 d after the medium was changed to natural seawater, there were no significant differences in the concentrations between the two biofilms.” Are data of two groups at 0 d, 6 d, 8 d, and 10 d statistically different from each other?
- Lines 378-380, “A possible reason for the appearance of quartz after incubation was that silicon dioxide was eluted first and then recrystallized as quartz during the incubation process”. Any scientific support/citation for this presumption?
- The manuscript needs to be carefully edited and formatted;
Minor:
- Line 57, “However, compost containing humicc substances”. What is humicc?
- Line 69, “structure where The biofilm is dotted ……” shall be “structure where the biofilm is dotted ……”;
- Caption for (B) in Figure 6 is missed (was typed in lines 286-287?).
- Line 314, “The closest strain to TO7 was”. It seems it is an incomplete sentence;
- It looks like Tables S1, S2, S3, and S4 are screenshot images, please re-create the table as some headers are not shown completely.
Author Response
Reviewer 1
Comments and Suggestions for Authors
This is an interesting study. The author described an approach to improve the steel slag usage as a material for the construction of seaweed forests. There are several concerns need to be addressed before publication.
(response)
Thank you for your meaningful comments. We modified and improved our manuscript according to your advice.
Major Concerns:
- A proper comparison with related work in the Introduction section is necessary;
(response)
We added some references about the relationships between steel slag and microbes to Introduction.
- In Figure 2, it was stated that “TO1 showed the highest contribution to iron elution”, but it should be noted that the error bars presented in Figure 2 are significant. Are the presented data statistically different from each other?
(response)
We added the statistical analyses to the data in Figure 2.
- What those marks X, Δ and О stand for in Table S1?
(response)
We described the meanings of these marks in Table S1.
- Proper descriptive phrase for section 3.2 is required
(response)
We replaced the title of section 3.2 to new appropriate one.
- It seems the iron concentration eluted by TO1A is less than that eluted by the control in Figure 4, why? Similarly, are each group statistically different from each other?
(response)
We added the statistically analyses to Figure 4. Only TO1 was significantly contributed to the elution of iron from CSMS.
- XRD was utilized to investigate the changes in mineral composition after incubation with TO1 culture, why is this composition change important?
(response)
The steel slag partially contains calcium carbonate and iron oxide minerals. For example, calcite, aragonite and vaterite are the crystal polymorph of CaCO3. However, the solubility of calcite, aragonite or vaterite in the water is different. The change of mineral composition in the steel slag indicated that the solubility of mineral components may be changed by the effects of bacteria. We added this explanation to Discussion.
- It is better for the authors to explain the mechanism of petal-like patterns formation in Figure 6;
(response)
Vaterite crystal tends to make the petal like patters on the crystal surface because the stability of vaterite crystal is unstable and repeats the precipitation and dissolution at the molecular level. Such unstable phase of vaterite may be a main factor to form the petal-like patterns. We added these explanations to Discussion.
- Lines 345-346, “At 4 d after the medium was changed to natural seawater, there were no significant differences in the concentrations between the two biofilms.” Are data of two groups at 0 d, 6 d, 8 d, and 10 d statistically different from each other?
(response)
We added the statistically analyses to Figure 9.
- Lines 378-380, “A possible reason for the appearance of quartz after incubation was that silicon dioxide was eluted first and then recrystallized as quartz during the incubation process”. Any scientific support/citation for this presumption?
(response)
We made this hypothesis about the formation of quartz from the results of XRD measurements and SEM observation. There are no references about this hypothesis because nobody measured the mineral change of steel slag before and after treatment of bacteria. We emphasize that the assumption about the formation of quartz is one of the hypothesis in the manuscript of Discussion.
- The manuscript needs to be carefully edited and formatted;
(response)
Thank you for your comments. We checked the format of journal and improved the whole manuscript.
Minor:
- Line 57, “However, compost containing humicc substances”. What is humicc?
(response)
We corrected it.
- Line 69, “structure where The biofilm is dotted ……” shall be “structure where the biofilm is dotted ……”;
(response)
We corrected it.
- Caption for (B) in Figure 6 is missed (was typed in lines 286-287?).
(response)
We are sorry that sentence structure was disordered by the inserted figure image. We corrected it.
- Line 314, “The closest strain to TO7 was”. It seems it is an incomplete sentence;
(response)
We are sorry that the inserted Figure 8 and its figure legend are confusing the sentence structure. We corrected it.
- It looks like Tables S1, S2, S3, and S4 are screenshot images, please re-create the table as some headers are not shown completely.
(response)
We replaced Table S1, S2, S3 and S4 to new tables with appropriate headers.
Reviewer 2 Report
Interesting paper with sound description and conclusions. It lacks however a correct characterization of the used slag (major review), in terms of:
- what type of steelmaking slag is ? From electric arc furnace? from ladle furnace? From converter? The exact origin of the slag in terms of steelmaking operation and technology should be explained and described;
- - physical and chemical characterization: I suggest to present a table with the chemical composition of the slag, a XRD spectra (those after reaction in water are well, but for the original material is also necessary, even for the seek of comparison) and eventually some electron microscopy presenting the internal microstructure of the slag and which phases is it composed.
I suggest also, as a minor revision, to indicate the sea water composition (page 2) in a table. And then, in page 3, you will not need to repeat it, just to mention the table.
Author Response
Reviewer 2
Comments and Suggestions for Authors
Interesting paper with sound description and conclusions. It lacks however a correct characterization of the used slag (major review), in terms of:
(response)
Thank you for your meaningful comments and suggestions. We modified and improved our manuscript according to your advice.
- what type of steelmaking slag is ? From electric arc furnace? from ladle furnace? From converter? The exact origin of the slag in terms of steelmaking operation and technology should be explained and described;
(response)
ISS was prepared from converters after optimizing refining process at Nagoya steel works of Nippon Steel Co., Ltd. We added this sentence to Materials and Methods
- - physical and chemical characterization: I suggest to present a table with the chemical composition of the slag, a XRD spectra (those after reaction in water are well, but for the original material is also necessary, even for the seek of comparison) and eventually some electron microscopy presenting the internal microstructure of the slag and which phases is it composed.
(response)
We added the XRD data and SEM observation of intact CSMS to Figure S3. The detailed chemical compositions of CMSM and SSHM are private information in Nippon Steel Co., Ltd. We are sorry that we can not show the exact chemical composition as an open resource.
I suggest also, as a minor revision, to indicate the sea water composition (page 2) in a table. And then, in page 3, you will not need to repeat it, just to mention the table.
(response)
We prepared Table S1 to show the components of Marine Broth 2216.
Round 2
Reviewer 1 Report
The authors had responded to my comments sufficiently and appropriately.
Reviewer 2 Report
No more comments.